# Wnt-Independent and Wnt-Dependent Effects of APC Loss on the Chemotherapeutic Response

**DOI:** 10.3390/ijms21217844

**Published:** 2020-10-22

**Authors:** Casey D. Stefanski, Jenifer R. Prosperi

**Affiliations:** 1Department of Biological Sciences, University of Notre Dame, Notre Dame, IN 46617, USA; cstefans@nd.edu; 2Mike and Josie Harper Cancer Research Institute, South Bend, IN 46617, USA; 3Department of Biochemistry and Molecular Biology, Indiana University School of Medicine-South Bend, South Bend, IN 46617, USA

**Keywords:** adenomatous polyposis coli, chemoresistance, WNT signaling

## Abstract

Resistance to chemotherapy occurs through mechanisms within the epithelial tumor cells or through interactions with components of the tumor microenvironment (TME). Chemoresistance and the development of recurrent tumors are two of the leading factors of cancer-related deaths. The Adenomatous Polyposis Coli (APC) tumor suppressor is lost in many different cancers, including colorectal, breast, and prostate cancer, and its loss correlates with a decreased overall survival in cancer patients. While APC is commonly known for its role as a negative regulator of the WNT pathway, APC has numerous binding partners and functional roles. Through APC’s interactions with DNA repair proteins, DNA replication proteins, tubulin, and other components, recent evidence has shown that APC regulates the chemotherapy response in cancer cells. In this review article, we provide an overview of some of the cellular processes in which APC participates and how they impact chemoresistance through both epithelial- and TME-derived mechanisms.

## 1. Introduction

Chemotherapy is the standard of care for most cancer types. The cytotoxic agents involved have different modes of action, including the disruption of DNA processes, microtubule networks, metabolism, and major signaling pathways. There are different classes of chemotherapeutics: alkylating agents; antimetabolites; anti-tumor antibiotics; topoisomerase inhibitors; mitotic inhibitors; and corticosteroids. There are also targeted therapies, hormone therapies, and immunotherapies. While the use of chemotherapeutic drugs demonstrated success in the clinic, patients often develop resistance to these treatment options [1]. Resistance can be classified as either intrinsic or acquired. Intrinsic resistance occurs when tumors exhibit resistance to the initial treatment and acquired resistance occurs after exposure to treatment [1,2]. Cancer cells have developed diverse methods to combat chemotherapy, depending on the drug’s mode of action. 

Genetic and epigenetic changes that alter drug influx and efflux, DNA repair, metabolism, cancer stemness, cell–cell interactions, and apoptosis cause a decreased drug response in patients. As described in this review article, the tumor suppressor Adenomatous Polyposis Coli (APC) influences a number of these drug-resistant phenotypes. We will discuss the use of APC as both a marker for the development of drug resistance and as a potential therapeutic target. Exploring the numerous processes in which APC participates will guide how the loss of APC can contribute to a decreased drug sensitivity. Here, we demonstrate that APC affects numerous chemoresistant pathways, altering responses to different classes of chemotherapeutic agents through epithelial- and tumor microenvironment (TME)-derived mechanisms.

## 2. Adenomatous Polyposis Coli and Cancer

*APC* is located on the long arm of chromosome 5. In the 1990s, APC was identified as the cause of familial adenomatous polyposis (FAP), which is a heritable disorder that causes numerous adenomas throughout the intestinal tract [3]. It was discovered that nonsense mutations in the *APC* gene resulting in a truncated protein were the driver of FAP. FAP patients present with hundreds of adenomatous polyps in their colon and many of these polyps will develop into cancer, providing an early demonstration of the role of APC in cancer [3,4]. 

In colorectal cancer (CRC), the dominant driver is the mutation of *APC*, which often occurs in the mutation cluster region (MCR), resulting in a truncated protein. Truncated APC demonstrates alternative functions, which will not be discussed in this review, but have recently been reviewed by Shay et al. [5]. Up to 90% of colon cancer cases have a mutation in *APC*, making it an important target in colon cancer therapeutics [6,7]. APC loss through mutation or promoter hypermethylation occurs in many different cancer types, including colon, prostate, breast, and non-small cell lung cancers [8,9,10,11,12]. The loss of APC through the upregulation of miR-135 contributes to colorectal, breast, and gastric cancer development [13,14,15]. This makes APC a viable therapeutic target in these cancer types. In addition, it was previously shown that the loss of APC resulted in a decrease in overall survival in non-small cell lung cancer (NSCLC) and breast cancer [9,16]. This not only demonstrates that the loss of APC could serve as a therapeutic marker, but also lends the question of why patients who present with APC-deficient tumors have a worse prognosis than APC-competent tumors. 

## 3. APC Affects Multiple Cellular Processes

APC is a large (310 kDa), multifunctional protein affecting many cellular processes, including proliferation, migration, DNA repair, and chromosomal segregation [17]. APC contains multiple functional domains, as determined by their different binding partners, including the oligomerization domain, armadillo repeats, 15 amino acid repeats, 20 amino acid repeats, SAMP (amino acids: serine, alanine, methionine, proline) repeats, the basic domain, the EB1-binding domain, and the PDZ-binding domain. In this review, we will discuss some of the most important interactions regulated by APC affecting chemosensitivity. 

One of the most studied roles of APC is its regulation of the Wnt/β-catenin signaling pathway. APC is a member of the β-catenin destruction complex, along with glycogen synthase kinase 3β (GSK3β), Axin, and casein kinase 1 (CK1), which allows for β-catenin to be targeted for proteasomal degradation. Without functional APC, β-catenin translocates into the nucleus and binds to a member of the TCF/LEF transcription factor family, activating the expression of many genes, including the S-phase regulators *c-myc* and *cyclin D1*, in order to promote proliferation. Mutations in *APC* causing dysregulated Wnt signaling are the primary mechanism for hyperproliferation in CRC, where Wnt signaling is essential in intestinal stem cell maintenance. Stem cells display an increased expression of Wnt proteins, but the essential role of Wnt in stem cells was first shown when TCF4 loss depleted the stem cell population [18]. In addition, Sato et al. made the remarkable discovery that Lgr5+ intestinal stem cells can organize into crypt-villus structures, which recapitulate the intestinal structure [19]. Uncontrolled Wnt signaling in granulocyte-macrophage progenitor cells enhanced the self-renewal and leukemic potential [20]. In addition to the multiple findings of Wnt directly impacting stem cells and cancer stem cells (CSCs), we made the novel observation that the loss of APC results in an increase in stem cells, independent of Wnt activation [21]. Our lab previously created cells from primary mammary tumors isolated from a mouse mammary tumor virus-polyoma middle T (MMTV-PyMT) transgenic mouse crossed with an *Apc^Min/+^* mouse (further referred to as MMTV-PyMT;*Apc^Min/+^* cells). In this model, we showed that there was no activation of the Wnt pathway and no loss of heterozygosity of APC. These two observations suggest that even minor changes to APC expression have a significant impact on the tumor phenotype. Using this model, we showed that APC loss resulted in an increase of aldehyde dehydrogenase (ALDH)-high cells compared to control cells. We have also shown enhanced mammosphere development in a model of APC-knockdown human breast cancer cells (unpublished results). Overall, these data suggest that APC regulates stem cell proliferation. 

In addition to Wnt-mediated cell cycle control, APC can also regulate cell cycle progression through the G2/M transition via interactions with Topoisomerase IIα, which is a regulator of the G2/M checkpoint. This interaction is essential in maintaining euploidy, where the loss of APC is deleterious to the chromosomal integrity [22,23,24]. Genomic instability is a fundamental hallmark of cancer and the role of APC in stabilizing microtubules (MTs) is essential to maintaining chromosomal stability. The loss of APC increases errors in mitotic spindle formation and chromosomal segregation [24,25,26]. APC is in the class of MT-associated proteins known as plus end-binding proteins (+TIPs) that regulate MT plus end dynamics, and interacts with the fellow +TIP, end-binding protein (EB1), which is important in maintaining proper chromosome alignment [27]. It was previously shown that MTs bound with APC exhibited increased growth and decreased transition states between growth and shortening [28]. The C-terminus of APC binds MTs, and truncations of APC in cancer suggest a loss of APC–MT interactions and a decreased MT stability, which promote tumor development.

In addition to the interaction between APC and MTs, recent evidence that APC also binds directly to actin suggests that APC could act as a regulator between MT dynamics and actin-based protrusions [29]. APC-deficient adenomas were previously shown to exhibit nondirected cell migration along the crypt-villus axis [30]. In addition, APC has been shown to interact with the guanine nucleotide exchange factor (GEF) known as Asef, stimulating Rac1 activation, membrane ruffling, lamellipodia formation, and cell migration [31,32]. The mouse mammary carcinoma cells referred to as 4T07 showed APC/β-catenin complexes at membrane protrusions. APC knockdown in the 4T07 cells resulted in decreased cell migration that was independent of Wnt pathway activation [33]. In a model of normal epithelium using MDCK cells, our lab showed that APC regulates cell migration through increased β1 integrin expression [34]. Together, this demonstrates APC’s function in maintaining cellular motility, with APC as a mediator between actin-microtubule crosstalk and allowing directional migration, indicating that APC also has a vital role in migration, which is an early step in metastasis [35,36]. The occurrence of metastasis, which is characterized by invasion and motility, is a leading culprit of cancer-related deaths. 

APC can either block or enhance DNA repair, depending on the type of DNA damage, the repair pathway involved, and the stage of tumorigenesis. APC’s role in DNA repair was first identified in long patch base excision repair (LP-BER), which is important in repairing abasic DNA damage. APC binds to DNA polymerase β (Pol-β), Flap endonuclease 1 (Fen-1), and AP endonuclease 1 (APE1), blocking LP-BER. In addition, APC interacts with replication protein A 32 (RPA32) during replication stress to promote ataxia telangiectasia and Rad3-related protein (ATR)-dependent phosphorylation of RPA32, checkpoint kinase 1 (Chk1), and phosphorylated histone H2AX (γ-H2AX), thereby regulating cell cycle reentry [37]. APC also associates with the DNA-dependent protein kinase catalytic subunits (DNA-PKcs) at the chromatin in response to double-stranded breaks, regulating non-homologous end joining (NHEJ) repair [38]. Using primary mammary tumor cells isolated from a MMTV-PyMT transgenic mouse crossed with an *Apc^Min/+^* mouse, we made the novel observation that the loss of APC in MMTV-PyMT;*Apc^Min/+^* mouse mammary cells decreased doxorubicin-induced DNA double-stranded breaks compared to wild-type specimens [39]. These roles of APC in different DNA repair pathways can not only promote tumorigenesis by increasing the genomic instability, but also have implications for DNA damaging anti-cancer agents. 

Overall, APC interacts with numerous binding partners affecting cellular processes that are required to maintain cellular homeostasis. While APC’s role in contributing to cancer development is well-established, more recent data have shown that the APC status may also influence cancer treatment.

## 4. APC Loss and Epithelial-Derived Chemoresistance

The development of resistance to standard chemotherapies is inevitable. Intracellular factors (i.e., efflux pumps or anti-apoptotic proteins) were originally believed to be the primary way in which cancer cells survived treatment (Figure 1). One of the most notable ways that chemoresistance occurs is through the overexpression of ATP binding cassette (ABC) transporters. The ABC transporters are membrane export pumps that efflux chemotherapeutics out of the cancer cell, thereby preventing drug-induced apoptosis. These transporters include the well-known pumps multidrug resistance 1 (MDR1) and multidrug resistance-associated protein 1 (MRP1). The expression of the MDR1 transporter is increased when *APC* is mutated [21,40], which may be a result of the Wnt/β-catenin signaling pathway, given that the *MDR1* gene promoter contains multiple binding elements for β-catenin/TCF4 [40,41]. The inhibition of Wnt signaling and suppression of MDR1 by overexpressing miR-506 re-sensitized CRC cells to oxaliplatin [42]. Silencing the long non-coding ribonucleic acid-homeobox transcript antisense ribonucleic acid (lncRNA-HOTAIR) in NSCLC cells also inhibited Wnt signaling, decreased the expression of MDR1 and MRP1, and decreased the cisplatin sensitivity [43]. Similarly, overexpressing β-catenin in oral squamous cell carcinoma cells caused cisplatin resistance by increasing the expression of MDR1 and MRP1 [44]. In addition to APC working through the Wnt pathway to increase MDR1, our lab has shown that the loss of APC can function independently of the Wnt pathway, through upregulation of the signal transducer and activator of transcription 3 (STAT3), to induce MDR1 expression [21,45]. Therefore, we and others have demonstrated that APC loss may influence chemoresistance by altering MDR1 through both Wnt-dependent and -independent mechanisms. While these ABC transporters have been shown to contribute to resistance, they unfortunately have not shown much success as therapeutic targets to prevent resistance in the clinic. Toxicity and interactions with other drugs used by the patients were found to be issues in clinical trials. Future drug development for ABC transport modulators includes improving drug delivery and preventing compensation from other drug exporters [46]. 

Many chemotherapies work by causing DNA damage and subsequent cell death; however, cancer cells have developed enhanced DNA repair pathways to repair the therapy-induced damage and avoid apoptosis. Depending on the type of damage inflicted, repair occurs through different signaling pathways. Oxidation, deamination, and alkylation damage are repaired via BER. 5-fluorouracil (5-FU) is a structural analog of uracil and thymine, and when uracil is processed, it creates an abasic site that is repaired via BER. Using CRC cells with different *APC* expression levels, it was shown that knocking down *APC* in HCT-116 cells, which have WT *APC*, resulted in resistance to 5-FU. In a complementary experiment, introducing full-length APC in LOVO cells, which have mutant *APC*, resulted in sensitivity to 5-FU [47,48]. It was also shown that APC regulation of the 5-FU therapeutic response stemmed from APC’s role in LP-BER [49]. Recently, the development of poly (ADP-Ribose) polymerase (PARP) inhibitors for treatment in BRCA-mutated cancers has highlighted the potential targeting of DNA repair proteins as a therapeutic approach. PARP inhibitors have improved patient prognoses; however, it was recently shown that Wnt signaling can regulate the response to PARP inhibitors in ovarian cancer [50,51]. Furthermore, it was found that *APC* was necessary to induce BER in metastatic breast cancer stem cells (mBCSCs) following treatment with the PARP inhibitor ABT-888 in mBCSCs pre-treated with the small molecule Quinacrine. While the knockdown of *APC* in mBCSCs decreased DNA damage, increased BER activity, and reduced apoptosis, the overexpression of *APC* in BT20 breast cancer cells displayed the opposite affects [52]. The double-stranded breaks that are created by chemotherapeutic agents are repaired through either NHEJ or homologous recombination (HR), in which APC is known to interact with RPA32 and DNA-PKcs. It was also shown that inhibitors for the DNA repair kinases ataxia telangiectasia mutated (ATM) and DNA-PK restored doxorubicin-induced apoptosis in doxorubicin-resistant APC-deficient breast cancer cells [39]. The role of APC in multiple DNA repair pathways demonstrates APC’s potential as a therapeutic marker like BRCA. 

The recognition that altered metabolic pathways in cancer cells contribute to chemoresistance was only established within the last decade. [53]. Cancer cells prefer aerobic glycolysis, referred to as the Warburg effect, partially because of mitochondrial dysfunction disrupting pyruvate metabolism. It was recently shown that APC regulated the expression of the *mitochondrial pyruvate carrier (MPC)*, affecting pyruvate metabolism and promoting tumor development [54,55]. Inhibiting MPC activity using UK5099 reduced the sensitivity to cisplatin compared to untreated LNCaP prostate cancer cells by inducing a stem-like phenotype [56]. These studies suggest that APC could work through MPC to alter the response to chemotherapy. In addition to MPC, glutathione metabolism is an important contributor to cisplatin resistance in NSCLC cells [57,58]. Glutathione is necessary in detoxification systems that combat reactive oxygen species (ROS) production induced by chemotherapeutic drugs. Cancer cells have increased detoxification and antioxidant systems to combat this ROS, such as glutathione S-transferase (GST), glutathione peroxidase (GSH-PX), and superoxide dismutase (SOD) [59]. Giera et al. demonstrated that the constitutive activation of β-catenin increased GST isoforms in mouse hepatomas, which can conjugate a myriad of hydrophobic and electrophilic molecules to glutathione, allowing for drug exportation [60,61]. It was also shown that glutamine activated the Wnt pathway and increased the expression of GSH-PX and SOD in Alzheimer’s disease. Inhibition of the Wnt pathway prevented these glutamine-induced antioxidants [62]. These Wnt-dependent processes downstream of glutathione are important because the utilization of glutathione synthesis inhibitors increases the sensitivity to cisplatin in cervical cancer [63]. Therefore, metabolic changes preventing chemotherapy efficacy represent another target for combination therapy for improving patient prognoses. Taken together, these studies suggest that APC’s role in Wnt regulation could contribute to resistance through alterations in glutathione-dependent detoxification systems; targeting these chemoresistant mechanisms could be a therapeutic target in APC-deficient cancers. 

Cancer stems cells (CSCs) are naturally more resistant to chemotherapy due to their slower cycling time, increased expression of ABC transporters, and enhanced DNA repair [64]. Inhibiting Wnt signaling decreased CSC marker proteins while increasing the sensitivity to 5-FU in CRC cells [65]. In addition, chemotherapeutics can induce cellular senescence and stemness via Wnt activation [66]. CSCs also produce less ROS while increasing antioxidant systems to combat any oxidative stress. Decreasing ROS scavenging in CSCs increased the sensitivity to radiation therapy [67,68]. Chemotherapy-resistant MMTV-PyMT;*Apc^Min/+^* cells lacking elevated Wnt signaling displayed increased numbers of ALDH-high cells, which is a marker of CSCs [21] and correlates with chemoresistance [69]. This suggests that APC has functions supporting stemness, independent of β-catenin. The use of an ALDH inhibitor increased the sensitivity of chemotherapy and radiotherapy in ALDH^high^/CD44^+^ breast cancer cells [70]. Therefore, APC regulation of CSCs may provide information on how to eliminate the chemoresistant CSC population.

The taxanes, which are a type of mitotic inhibitor, stabilize microtubules (MTs), disrupting mitosis and cell cycle progression. Directly preventing taxane’s mode of action by preventing the taxane stabilization of MTs is one mechanism leading to taxane resistance, which can be accomplished through alterations to the MT network, including microtubule-associated proteins (MAPs) such as APC. APC’s role in stabilizing MTs, independent of its interaction with the plus end MT binding protein EB1, can regulate the cell’s response to paclitaxel [71]. Low doses of paclitaxel in APC-deficient intestinal enterocytes failed to decrease the number of mitotic cells compared to wild-type enterocytes [72]. Resistance to paclitaxel was also seen in breast cells lacking APC [21]. APC loss decreases the efficacy of microtubule-stabilizing agents; however, the use of the MT-destabilizing agent vinorelbine preferentially killed APC-deficient U2OS cells through its ability to induce apoptosis during interphase, as well as mitosis [73]. This demonstrates the need to consider the molecular tumor profile to establish an appropriate therapeutic approach. 

The ultimate goal of chemotherapy is to induce apoptosis in cancer cells. Therefore, the mis-regulation of pro- and anti-apoptosis proteins can induce chemoresistance. Bladder cancer patients who were resistant to cisplatin treatment also showed an increased nuclear expression of survivin, which is a Wnt target that is an inhibitor of the apoptosis protein (IAP) [74]. The co-expression of APC and AXIN employed to inhibit Wnt signaling reduced survivin expression and inhibited cell growth [75]. As survivin is primarily expressed in cancer cells, inhibitors are currently being evaluated as a single agent or in combination [76,77]. Another family of proteins regulating apoptosis are the B-cell lymphoma 2 (BCL-2) family proteins, which are crucial in the mitochondrial death pathway. The loss of APC increased the expression of the pro-survival protein BCL-2, but not MCL-1, in MMTV-PyMT;*Apc^Min/+^* cells [45]. BCL-2 inhibitors are clinically being investigated as a potential combination therapy. Venclexta, which is a BCL-2 small molecule inhibitor and FDA approved for treatment in lymphocytic leukemia, is therapeutically efficacious in refractory and relapsed chronic lymphocytic leukemia, including in combating resistance [78]. An increased expression of the downstream apoptotic mediators caspase 3, 7, and 9 was found in *Apc*-mutated colon cancer [79], suggesting that the loss of APC increases the apoptotic response. This demonstrates that APC regulates the expression of caspases and other apoptotic proteins. 

The heterogeneous nature of tumors also contributes to chemotherapy resistance through cell–cell interactions. Wnt activation through miR-103/107 targeting Axin2 promoted stemness and chemoresistance [80]. MiR-130a induced cisplatin resistance in hepatocellular carcinoma cells (HCCs) by inhibiting the tumor suppressor gene RUNX3, thereby activating Wnt [81]. In addition, miR-92a is upregulated in 5-FU-resistant CRC cells, and an ectopic expression of miR-92a induced chemoresistance in CRC cells through increased Wnt activation by targeting negative regulators of Wnt signaling. Interleukin-6 (IL-6)/STAT3 was found to increase miR-92a expression, promoting a stem-like phenotype to induce resistance [82]. Our lab demonstrated that doxorubicin resistance in MMTV-PyMT;*Apc^Min/+^* cells was mediated by increased STAT3 expression, independent of the Wnt pathway [45]. The upregulation of miR-135 induced paclitaxel resistance via downregulating APC in lung, uterine, and breast cancer cells [83,84]. Recently, the re-expression of APC was shown using the lncRNA SMAD5-AS1, which acts as a competitive endogenous RNA to miR-135, preventing APC inhibition, suggesting the use of lncRNAs for restoring gene expression [85]. APC loss mediates chemoresistance through cell–cell interactions via Wnt-dependent and -independent mechanisms.

## 5. APC Loss and Tumor Microenvironment-Derived Chemoresistance

Within the past decade, the shifted emphasis onto the tumor microenvironment (TME) has demonstrated that extracellular factors support the development of chemoresistance [86]. The heterogeneous tumor landscape, which is comprised of both cellular components (cancer, endothelial, and stromal cells) and noncellular components (the extracellular matrix (ECM) and soluble factors), creates barriers that promote chemoresistance. Cancer cells can influence the TME to support their own survival. Cells that are normally anti-tumorigenic cells, including fibroblasts and macrophages, can be converted into tumor-supporting cells, such as cancer-associated fibroblasts (CAFs) and tumor-associated macrophages (TAMs) [86]. Exosomes, which are small lipid vesicles containing proteins and genetic material, are secreted by different types of cells into the TME, inducing pro-invasion and survival signaling in cancer cells [87]. In addition, inflammatory factors in the TME mediate epithelial–mesenchymal transition (EMT), metastasis, and resistance through the expansion and recruitment of CAFs and immune cells such as macrophages [88]. Together, the TME and tumor cells crosstalk to support cancer cell survival (Figure 2). 

A defining feature of the tumor landscape is pockets of hypoxia, demanding increased vascular formation or angiogenesis to support the metabolic need of cancer cells. Many drugs induce cell death through the production of free radicals and oxidative stress and therefore require oxygen to induce apoptosis. Hypoxic glioblastoma cells released exosomes containing miR-301a into the TME, activating Wnt signaling and decreasing the sensitivity to radiation [89]. In addition, APC has an antagonistic relationship with hypoxia-inducible factor-1α (HIF-1α), where HIF-1α represses the transcription of APC and APC loss increases HIF-1α [90]. In a hypoxic model of diffuse large B-cell lymphoma, a reduced HIF-1α expression resulted in enhanced doxorubicin-induced apoptosis [91]. HIF-1α inhibition also increased the chemosensitivity under hypoxic conditions in acute lymphocytic leukemia cells [92]. Reduced angiogenesis to tumors impeded drug delivery, as well as free radical formation from oxygen depletion. APC binds to the Rac-specific GEF Asef, enhancing Asef activity [32]. The APC/Asef complex increased endothelial cell migration and tube formation through increased basic fibroblast growth factor (bFGF) and vascular endothelial growth factor (VEGF). Asef^−/−^ mice also displayed reduced microvessel formation [93]. This suggests that APC/Asef could be necessary for angiogenesis to supply tumor cells with an adequate blood supply. 

Mesenchymal stromal/stem cells (MSCs) are recruited to tumors and release soluble factors to the cancer cells [86]. Human umbilical cord-derived MSCs co-cultured with a human cholangiocarcinoma cell line produced conditioned media that contributed to drug resistance and metastasis through increased Wnt activation and the upregulation of Wnt target genes (matrix metalloproteinase 2 (MMP2), cyclin D1, and c-myc) [94]. Irradiated MSCs supported cancer stemness in HCCs via Wnt activation, contributing to radiation resistance [95]. Moreover, FAP patients often present with desmoid tumors that originate from mesenchymal cells because APC loss dysregulates Wnt signaling and increases growth in MSCs [96]. Therefore, APC regulates the growth of MSCs, which contribute to a chemoresistant phenotype. Interestingly, some studies have suggested that MSCs are differentiated into CAFs, but this is still debated [86].

Some of the most studied aspects of the TME contributing to resistance are CAFs and their transport and the release of soluble factors. Exosomes secreted from CAFs released Wnt ligands into the TME, which activated Wnt signaling and drug resistance in differentiated CRC cells. In contrast, inhibiting Wnt secretion in Wnt3a overexpressing CAFs prevented resistance [97]. CAFs express the lncRNA colorectal cancer-associated lncRNA (CCAL), which activates Wnt signaling to promote oxaliplatin resistance in CRC cells [98]. CCAL-induced Wnt activation also increased MDR1 expression [99], which may allow for alterations in drug efflux. In CRC cells, CAFs transfer exosomes to increase miR-92a-3p, which activates Wnt signaling, preventing mitochondrial apoptosis and increasing stemness, EMT, and 5-FU/Oxaliplatin resistance [100]. Following chemotherapy or radiation, fibroblast-secreted WNT16B activated Wnt in primary prostate cells through NF-κB to promote a mesenchymal phenotype and evade apoptosis [101]. While these data point to a role for Wnt signaling, an inference could be made that one method of Wnt signaling activation is the loss of APC, which could lead to the activation of the same resistance mechanisms. The stromal depletion of APC in the uterine stroma induced a myofibroblast-like phenotype, similar to the phenotype of CAFs, contributing to endometrial hyperplasia and carcinogenesis through unopposed estrogen signaling in endometrial cells [102]. These studies demonstrate that an APC-deficient stroma can increase tumorgenicity and chemosensitivity. 

TAMs also secrete soluble factors into the TME that influence the therapeutic response [103]. Crosstalk between tumor cells and TAMs or M2-like macrophages promotes cancer cell survival. Not only do M2 macrophages secrete Wnt ligands to stimulate Wnt signaling in epithelial tumor cells, inducing stemness and the EMT phenotype, but Wnt6 expression in granulomatous lesions drives macrophage polarization toward an M2-like phenotype, which was also seen in HCCs [104,105]. HCCs were also shown to secrete Wnt ligands to activate M2 macrophages into TAMs [106]. This demonstrates a positive feedback loop between cells, where stromal Wnt drives cancer progression and cancer cell-derived Wnt stimulates a tumor-supporting microenvironment. TAM-secreted CCL5 promoted migration and EMT in prostate cancer cells, while also supporting CSC self-renewal by activating β-catenin/STAT3 [107]. In addition, TAMs release the cytokine IL-6, reducing the 5-FU response in CRC subcutaneous tumors via IL-6/STAT3 [108]. We found a loss of APC-induced doxorubicin resistance through an increased expression of STAT3 in MMTV-PyMT;*Apc^Min/+^* cells [45]. *Apc^Min/+^* mice have an increased expression of the M1 macrophage marker IL-23 and the M2 macrophage markers IL-13 and CCL17. The knock-out of monocyte chemoattractant protein 1 (MCP-1), which is an important chemokine for macrophage recruitment, decreased the expression of IL-1 and IL-6 in polyps of *Apc^Min/+^* mice. No change was observed in β-catenin staining between *Apc^Min/+^* and *Apc^Min/+^/*MCP-1^−/−^ mice, demonstrating that macrophage recruitment in *Apc^Min/+^* mice is independent of Wnt activation [109]. The loss of APC and subsequent Wnt signaling support macrophage infiltration into the tumor, which negatively impacts the drug response.

The newest promising therapy option in oncology is immunotherapy. Tumor cells can affect the TME by preventing immune cell recruitment to the tumor, rendering immunotherapy ineffective. Metastatic melanoma cells with active Wnt signaling displayed a decreased expression of the chemokine CCL4, preventing CD103^+^ dendritic cell (DC) recruitment and the subsequent activation of CD8^+^ cytotoxic T-cells. The direct injection of DCs increased the sensitivity to the immune checkpoint blockade [110,111]. This demonstrates that Wnt activation preventing DC recruitment can affect the response to immunotherapy. Similarly, Wnt signaling activation in HCCs reduced DC recruitment and the efficacy of the immune checkpoint blockade anti-PD-1 [112]. Wnt inhibitors are being evaluated, in combination with immune checkpoint inhibitors, such as anti-PDL-1, in preclinical trials [64]. *Apc*-deficient mice exhibited an impaired differentiation of T regulatory cells (Tregs) and high levels of the immune suppressive cytokine IL-10 because APC loss reduced the nuclear localization of Nuclear Factor of Activated T Cells (NFAT) [113]. This immune alteration was shown to occur partially outside of APC’s role in Wnt regulation. CD4^+^ T cell development was also impaired in *Apc^Min/+^* mice [114], demonstrating that APC loss alters the immune cell response, which may reduce the immunotherapy efficacy [115]. Overall, APC should be considered as its own therapeutic marker because of APC’s Wnt-dependent and -independent roles. 

## 6. Conclusions

APC is lost in numerous cancer types, through either mutations or promoter hypermethylation. More recent studies have shown that APC loss can cause resistance and may be used to predict the response to chemotherapy. The loss of APC and lack of control of the APC-mediated cellular functions present a therapeutic challenge given that the restoration of APC has been clinically challenging. Resistance can occur intracellularly, intercellularly (with TME), or through a relationship between the cells and the ECM. APC loss affects many cellular processes and may thus be responsible for multiple mechanisms of chemotherapy resistance. Future studies need to address whether APC can be used as a therapeutic marker. While the role of Wnt activation contributing to chemoresistance is well-studied, the Wnt-independent roles of APC are largely ignored. These Wnt-independent mechanisms demonstrate the importance of APC as a unique therapeutic marker outside of Wnt/β-catenin activation. Furthermore, the role of APC in the TME and how this affects the therapeutic response is understudied. Understanding how APC imparts chemoresistance will be imperative in discovering combination therapies to overcome resistance and improve patient outcomes. 

## Figures and Tables

**Figure 1 ijms-21-07844-f001:**
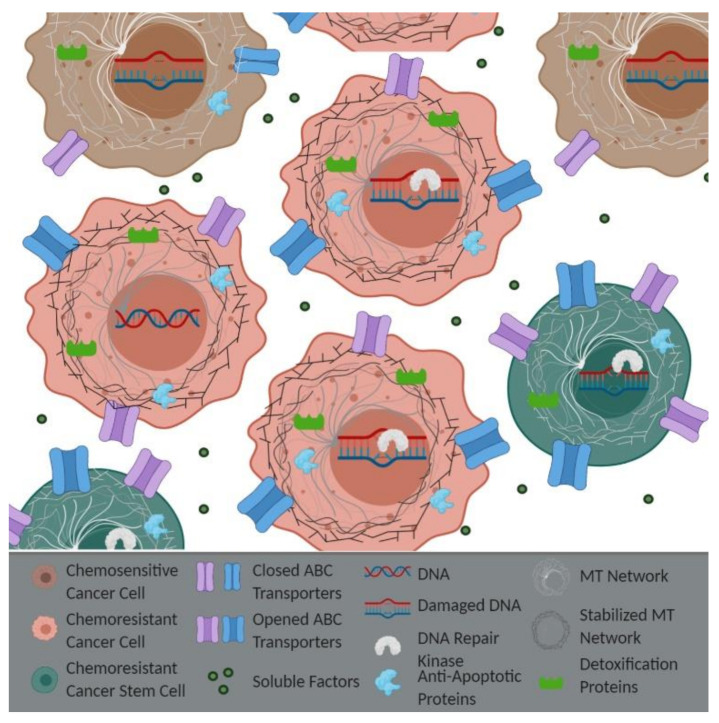
Epithelial-Derived Resistance: Intracellular resistance can occur though different mechanisms. Active (open) ATP binding cassette (ABC) transporters on the chemoresistant cells are responsible for the efflux of drugs. Decreased drug-induced DNA damage and enhanced DNA repair kinase expression and/or activity prevent the induction of apoptosis. An increased expression of anti-apoptotic proteins, alterations in detoxification systems, altered microtubule (MT) networks, and various cell–cell interactions can also contribute to drug resistance. Images made with BioRender.

**Figure 2 ijms-21-07844-f002:**
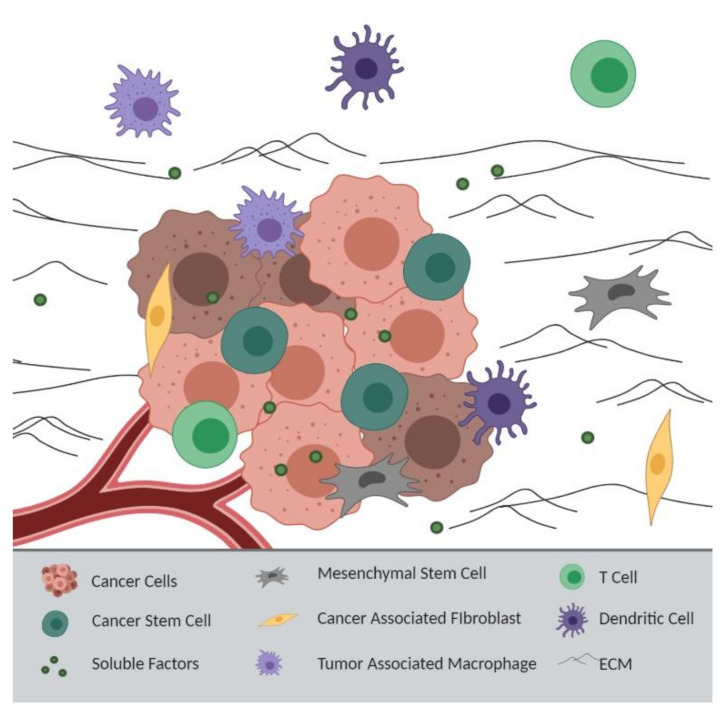
Tumor Microenvironment-Derived Resistance: Intercellular crosstalk between cancer cells and the tumor microenvironment (TME), including mesenchymal stem cells, cancer-associated fibroblasts, tumor-associated macrophages, T cells, and dendritic cells, promotes chemoresistance through the release of soluble factors. The extracellular matrix (ECM) can also act as a barrier to chemotherapeutics, and similarly, a limited blood supply to the tumor can prevent drug accumulation at the tumor site. Images made with BioRender.

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
