# Peer review of "Wnt-Independent and Wnt-Dependent Effects of APC Loss on the Chemotherapeutic Response"

_ijms, 2020, doi:10.3390/ijms21217844_

Round 1

Reviewer 1 Report

This is a scholarly written review giving a broad and apparently exhaustive overview of the role of APC and in more general the  Wnt pathway in resistance of cancers to chemotherapy. Sometimes the connection between APC and resistance is quite speculative, but these speculations remain within the limits of such a review.

There are only a few critical aspects, which the authors might consider for revision.

1 I found the figures provided not very informative as they show quite general content and do not relate directly to the main topic of the review.  Instead it would be good to summarize in a figure format the different molecular targets at which APC acts in chemoresistance, perhaps focusing on those with direct connection. Maybe this could include which of these effects are related to canonical Wnt signaling, and which are clearly not, which is often debated in the field.

2 Several evidence claiming a role of APC is in fact dealing with the role of the canonical Wnt pathway in general, for instance when it comes to Wnt secretion reported from different cells in the tumor microenvironment in chapter 4. These studies do not suggest a role of "loss of APC"  in chemotherapeutic response, but of the Wnt pathway in general. The authors might adapt the title of the review accordingly, by including Wnt signaling.

3 When they report about their own model on MMTV-PyMT/APCmin it was not explicitly stated that there is no LOH of APC so that it is not a complete loss of full-size APC that would explain their results.

4 line 63-68: Axin binding SAMP repeats should be mentioned as well.

5 line 194: It was not getting clear by which mechanism alterations in MPC can change chemotherapy response.

Author Response

We thank the reviewers for the thorough review of our paper, and the positive comments received. We have addressed the suggestions below and within the revised manuscript as indicated.

Reviewer 1

I found the figures provided not very informative as they show quite general content and do not relate directly to the main topic of the review. Instead it would be good to summarize in a figure format the different molecular targets at which APC acts in chemoresistance, perhaps focusing on those with direct connection. Maybe this could include which of these effects are related to canonical Wnt signaling, and which are clearly not, which is often debated in the field. We appreciate the comment. Our goal with the figures is for them to be descriptive of the potential epithelial or microenvironmental methods of developing chemoresistance, with this highlighting the pathways in which APC can be involved. In the text, we have discussed the way that APC (and sometimes the Wnt pathway) are able to influence these specific targets. While the idea of separating the topics based on Wnt dependence, or lack thereof, would be an excellent addition, it would be muddied by the fact that some pathways (ie, MDR1) are both Wnt-dependent and Wnt-independent based on the model system. We have also published a paper showing the multiple functions of APC depending on APC localization (Prosperi and Goss, 2011), which would also have to be considered.
Several evidence claiming a role of APC is in fact dealing with the role of the canonical Wnt pathway in general, for instance when it comes to Wnt secretion reported from different cells in the tumor microenvironment in chapter 4. These studies do not suggest a role of "loss of APC" in chemotherapeutic response, but of the Wnt pathway in general. The authors might adapt the title of the review accordingly, by including Wnt signaling. We have added text (line 327-30) to emphasize the fact that this is an inference about the potential role of APC loss, given it’s responsibility as a negative regulator of the Wnt signaling pathway. We’ve also reflected in the title that APC loss functions through Wnt-independent and Wnt-dependent mechanisms. While there certainly is more literature on Wnt pathway roles in chemotherapeutic response, including all of that information would be beyond the scope of this review article. We appreciate that the reviewer commented that we have speculations about the role of APC, but that this is within the limits of the review.
When they report about their own model on MMTV-PyMT/APCmin it was not explicitly stated that there is no LOH of APC so that it is not a complete loss of full-size APC that would explain their results. We appreciate the reviewer pointing out this important finding, as it suggests small alterations in APC expression can result in phenotypic changes. In line 87-89, we have added discussion about the lack of APC LOH, and the potential meaning of that finding.
line 63-68: Axin binding SAMP repeats should be mentioned as well. We apologize for the oversight. The SAMP repeats have been added (line 67).
line 194: It was not getting clear by which mechanism alterations in MPC can change chemotherapy response. This is now around line 198-201, and we have added text with information from reference 55, showing that MPC changes chemotherapy response through enhancement of a stem-like phenotype.

Reviewer 2

Line 152: our lab change with we (to much our lab in this part of the manuscript). Thank you. We have made the change (at line 155 – shifted with changes mentioned above).
Line 179: The sentence...PARP inhibitors have shown promise clinically...sounds weird. This is now lines 181-182, and we have changed it to “PARP inhibitors have improved patient prognosis.”
I would also suggest authors to look for some new, fresh reviews about EMT, microenvironment and drug resistance (reviews that target different types of cancer..not only specific one). We have added information at lines 281-283 and 341-343 to reflect on these changes.

Reviewer 2 Report

The manuscript submitted by Stefanski and Prosperi sounds interesting and I found it acceptable for publication in IJMS.

There are few minor things which need correction:

1) Line 152: our lab change with we (to much our lab in this part of the manuscript)

2) Line 179: The sentence...PARP inhibitors have shown promise clinically...sounds wierd

3) I would also suggest authors to look for some new, fresh reviews about EMT, microenvironment and drug resistance (reviews that target different types of cancer..not only specific one)

Author Response

(The authors gave the same response as above.)
